

# Sex difference in the association between triglyceride and intracerebral bleeding risk after intravenous thrombolysis for acute ischemic stroke, a multi-center retrospective study

Qilin Yuan[1,2,*], Ying Han[3,*], Shuangfang Fang[1,2], Hanhan Lei[1,2], Huapin Huang[1,2], Huiying Lin[1,2], Xiaomin Wu[1,2], Ronghua Chen[1,2], Zhiting Chen[1,2], Jie Chen[4], Hangfeng Li[5], Nan Liu[1,6] and Houwei Du[1,2]

[1] Department of Neurology, Fujian Medical University Union Hospital, Fuzhou, Fujian Province, China
[2] Institute of Clinical Neurology, Fujian Medical University, Fuzhou, Fujian Province, China
[3] Department of Geriatrics, Fujian Medical University Union Hospital, Fuzhou, Fujian Province, China
[4] Department of Neurology, Fujian Provincial Hospital South Branch, Fuzhou, Fujian Province, China
[5] Department of Neurology, Longyan First Affiliated Hospital of Fujian Medical University, Longyan, Fujian Province, China
[6] Department of Rehabilitation, Fujian Medical University Union Hospital, Fuzhou, Fujian Province, China
[*] These authors contributed equally to this work.

Corresponding authors
Nan Liu, xieheliunan1984@fjmu.edu.cn
Houwei Du, houweidu@fjmu.edu.cn

## ABSTRACT

**Background.** Whether the relationship of intracerebral bleeding risk with lipid profile may vary by sex remains unclear. This study aims to investigate potential sex differences in the association between lipid profile and the risk of symptomatic intracerebral hemorrhage (sICH) in patients with acute ischemic stroke (AIS) who received intravenous thrombolysis using recombinant tissue plasminogen activator (r-tPA).

**Methods.** This multicenter retrospective observational study analyzed patients with AIS treated with intravenous r-tPA. sICH was defined as a worsening of 4 or higher points in the National Institutes of Health Stroke Scale (NIHSS) score within 36 hours after intravenous thrombolysis in any hemorrhage subtype. We assessed the odds ratio (OR) with 95% confidence interval (CI) of lipid profile for sICH for each sex using logistic regression models adjusted for potential confounding factors.

**Results.** Of 957 participants (median age 68 (interquartile range, 59–75), men 628 (65.6%)), 56 sICH events (36 (5.7%) in men and 20 (6.1%) in women) were observed. The risk of sICH in men decreased with increasing serum levels of triglyceride after adjustment for confounding factors (*vs* lowest tertile, medium tertile OR 0.39, 95% CI [0.17–0.91], top tertile OR 0.33, 95% CI [0.13–0.84], overall $p = 0.021$; per point increase, adjusted OR 0.29, 95% CI [0.13–0.63], $p = 0.002$). Neither serum levels of total cholesterol nor low-density lipoprotein (LDL) was associated with sICH in men. In women, there was no association between any of the lipid levels and the risk of sICH.

**Conclusions.** This study indicated that the association between serum levels of triglyceride and sICH may vary by sex. In men, increased triglyceride levels decrease the risk of sICH; in women, this association was lost. Further studies on the biological mechanisms for sex differences in stroke risk associated with triglyceride are needed.

## BACKGROUND

Symptomatic intracerebral hemorrhage (sICH) accounts for the most feared complication after intravenous thrombolysis with recombinant tissue plasminogen activator (r-tPA) for acute ischemic stroke (AIS) (*Cordonnier et al., 2018*; *Powers et al., 2019*). Previous studies have identified different predictors of sICH after r-tPA thrombolysis, such as diabetes, hypertension, elevated serum glucose, and lower platelet count (*Nisar, Hanumanthu & Khandelwal, 2019*). Epidemiological studies have shown potential associations between lower cholesterol levels and the risk of hemorrhagic stroke. The Safe Implementation of Treatments in Stroke-International Stroke Thrombolysis Register (SITS-ISTR) study showed that among 35,314 patients, lower total cholesterol levels were associated with a higher risk of sICH (*Escudero-Martínez et al., 2023*). Data from Eastern Asia showed that decreasing serum cholesterol levels had trends towards a lower risk of non-hemorrhagic stroke (odds ratio (OR) for 0.6 mmol/L decrease, 0.77 [0.57–1.06]) and an increased odds of hemorrhagic stroke (1.27 [0.84–1.91]) (*Anonymous, 1998*). A large observational study of 45,079 AIS patients treated with intravenous alteplase showed that although there was no difference in the rate of sICH ($p = 0.13$), male sex was associated with a higher risk of sICH (OR 1.25, 95% CI [1.04–1.51], $p = 0.02$) after adjustment for confounding variables (*Lorenzano et al., 2013*). Several previous studies have indicated an association between low-density lipoprotein cholesterol (LDL) and symptomatic hemorrhagic transformation in a mixed population of patients undergoing IV thrombolysis, suggesting sex may influence the association between lipid profile and intracerebral bleeding risk (*Bang et al., 2007*; *Uyttenboogaart et al., 2008*). To our knowledge, sex differences in the relationship between lipid profile and the risk of sICH after intravenous r-tPA for AIS have not been fully understood. Knowledge of sex differences might be of importance in improving preventive and in-hospital treatment strategies in the AIS population. We aimed to investigate the association of the traditional harmful lipid profiles (including the evaluation of total cholesterol, low-density lipoprotein (LDL), and triglyceride) with the odds of sICH according to sex in this retrospective study.

## METHODS

### Study design and participants

We included consecutively intravenous thrombolyzed AIS patients from three teaching hospitals of Fujian Medical University between January 2013 and December 2022, in this multicenter, retrospective observational study. Eligibility criteria included age 18 years or older, diagnosis of AIS confirmed by head computed tomography or magnetic resonance imaging, and complete data on serum lipid profiles.

Information on demographics and clinical characteristics was collected from digital databases using a standardized data collection form. Hyperlipidemia was defined as (1)

serum cholesterol levels ≥4.7 mmol/L, (2) triglyceride levels ≥2.3 mmol/L, or (3) low-density lipoprotein levels ≥4.1 mmol/L (*Anonymous, 2016*). Hypertension was defined as reported systolic blood pressure (SBP) ≥ 140 mmHg or diastolic blood pressure ≥90 mmHg, based on the average of three blood pressure measurements, previous diagnosis of hypertension, patient's self-report of hypertension, or antihypertensive use. Diabetes, irrespective of type, was defined as a fasting blood glucose level ≥ 7.1 mmol/L twice, previous diagnosis of diabetes, or hypoglycemic use). Ischemic heart disease was defined as a history of myocardial infarction, angina, or coronary artery disease. Atrial fibrillation includes chronic or paroxysmal atrial fibrillation. Chronic heart failure was defined as a documented history or cardiologist's diagnosis (*Goldstein et al., 2006*). Trained clinicians assessed initial stroke severity using the National Institutes of Health Stroke Scale (NIHSS) score. Fasting lipid levels after stroke onset were measured on the morning after admission in each institution (*Cuadrado-Godia et al., 2009*). Stroke subtype was categorized based on The Trial of Org 10172 in Acute Stroke Treatment (TOAST) classification (*Gordon et al., 1993*).

## Outcomes

sICH was defined as a worsening of 4 or higher points in the NIHSS score due to intracerebral hemorrhage verified by a computed tomography (CT) scan within 36 h after thrombolysis (*Hacke et al., 2008*). One experienced neurologist who was blinded to the study design assessed the CT or MRI images and medical records to define sICH events.

## Missing data

Missing data were imputed using regression imputation with maximum likelihood estimates (*Haukoos & Newgard, 2007*).

## Statistical analysis

Categorical variables were compared using the Chi-Square test and the Mann–Whitney U tests where appropriate. For categorical variables, percentage proportions were calculated by dividing the number of events by the total number of patients. Mean with standard deviation (SD) or median with interquartile ranges (IQR) were calculated for normally distributed and non-normally continuous variables. In the current study, serum levels of total cholesterol, LDL, and triglyceride were analyzed as both ordinal categorical (tertiles, with the lowest tertile defined as reference) and continuous variables. Multivariable analysis was performed using stepwise logistic regression independent models for each lipid variable in men and women. Model 1 was adjusted for age and NIHSS; Model 2 was adjusted for all variables that showed an association in the univariate analysis with a $p < 0.1$. We did two sensitivity analyses to test the robustness of our findings: (i) excluding those who received statin treatment before index ischemic stroke and (ii) excluding those who underwent endovascular thrombectomy. We additionally conducted a moderation analysis to explore the role of sex in the association between lipid levels and sICH (*Ballarini et al., 2021*). Analyses were performed in the entire group as well as sex subgroups. All statistical analyses were done using the SPSS software (version 25.0) and R (version 4.5). A *p*-value of <0.05 was considered statistically significant.

### Ethics approval and consent to participate

The studies involving human participants were reviewed and approved by the Fujian Medical University Union Hospital Ethics Committees (NO. 2019KY076). Written informed consent for participation was waived in accordance with the national legislation and the institutional requirements.

## RESULTS

We included 957 participants (median age 68 [IQR 59–75], 628 men [65.6%]) in the final analysis (Fig. 1), with 56 sICH events (5.7% in men and 6.1% in women) observed. The median baseline NIHSS score was 6 [IQR 3–12].

Table 1 summarizes the baseline characteristics and univariate comparisons among male and female patients. Compared to female participants, male participants were younger (66 [58-74] *vs.* 70 [60-77], $p < 0.001$), more likely to be a current smoker (294 [46.8%] *vs.* 4 [1.2%], $p < 0.001$) and an often drinker (110 [17.5%] *vs.* 6 [1.8%], $p < 0.001$), more likely to have a previous stroke (100 [15.9%] *vs.* 34 [10.3%], $p = 0.018$) and had a less severe stroke (a NIHSS score of 6 [3–11] *vs.* 8 [4–14], $p = 0.002$). The proportion of atrial fibrillation in men was lower than in women (147 [23.4%] *vs.* 134 [40.7%], $p < 0.001$). The serum levels of total cholesterol (mmol/L, $4.48 \pm 1.16$ *vs.* $4.74 \pm 1.04$, $p < 0.001$) and LDL (mmol/L, $2.98 \pm 1.13$ mmol/L *vs.* $3.19 \pm 1.12$ mmol/L, $p = 0.005$) were lower in men than in women. The serum level of triglyceride was higher in men than in women, although the difference did not reach statistical significance (mmol/L, 1.29 [IQR 0.91–1.88] *vs.* 1.20 [0.87–1.75], $p = 0.065$). There were no significant difference between the sexes in terms of pre-stroke statin use (44 [7.8%] *vs.* 24 [8.1%], $p = 0.863$).

Baseline characteristics among patients with and without sICH events in men and women are shown in Table 2. In comparison, in men, the variables associated with sICH in the univariate analysis were increasing age ($p = 0.001$), atrial fibrillation ($p = 0.002$), hyperlipidemia ($p = 0.004$), increasing initial NIHSS score ($p < 0.001$), lower blood platelet count ($p = 0.005$), pre-thrombolysis blood glucose ($p = 0.001$), and serum levels of triglyceride ($p < 0.001$, Table 2). In comparison, in women, the variables associated with sICH in the univariate analysis were age ($p = 0.015$), diabetes ($p = 0.013$), atrial fibrillation ($p = 0.001$), initial NIHSS score ($p = 0.001$), and pre-thrombolysis blood glucose ($p = 0.004$). Female patients with and without sICH had similar serum levels of total cholesterol, LDL, and triglyceride. Stroke subtypes in patients with and without sICH were significantly different in both men and women ($p < 0.05$, respectively, Table 2).

Table 3 and Tables S1, S2 show the associations of lipid profile with the risk of sICH. Univariable regression analysis showed that serum level of triglyceride was inversely associated with the risk of sICH (*vs* lowest tertile, medium tertile OR 0.39, 95% CI [0.17–0.88], top tertile OR 0.29, 95% CI [0.12–0.71], $p$ for trend = 0.004, overall $p = 0.008$; per mmol/L increase, OR 0.26, 95% CI [0.12–0.56], $p = 0.001$) in men. This association was not found in women (*vs* lowest tertile, medium tertile OR 0.91, 95% CI [0.28–2.96], top tertile OR 1.49, 95% CI [0.52–4.25], p for trend = 0.464, overall $p = 0.646$; per mmol/L increase, OR 1.18, 95% CI [0.80–1.74], $p = 0.417$, Table 3). After adjustment for age and

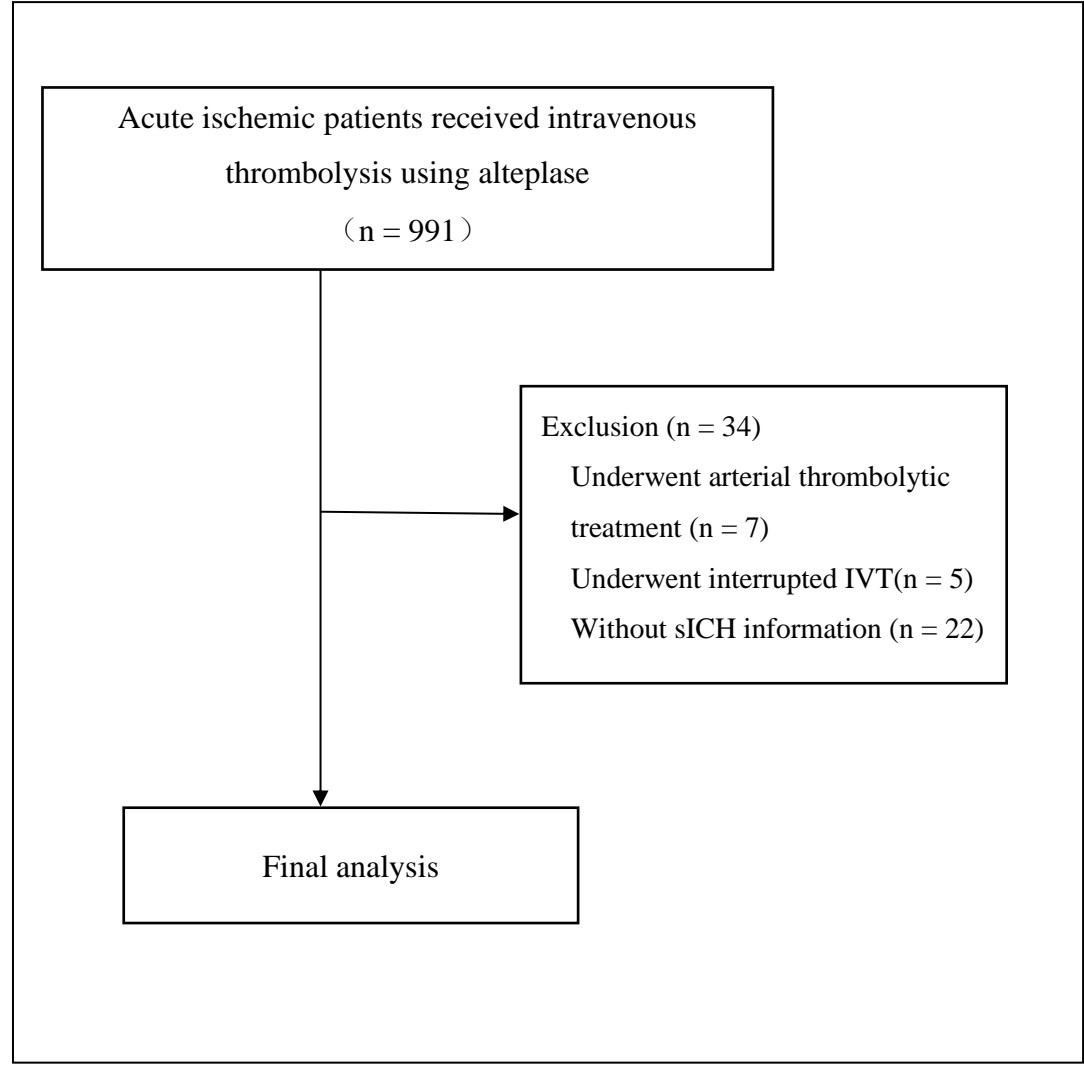

**Figure 1   Flow chart.**

NIHSS (model 1), a higher level of triglyceride was inversely associated with the risk of sICH in men, but not in women. This association remained in the multiple regression Model 2 in men (*vs* lowest tertile, medium tertile OR 0.39, 95% CI [0.17–0.91], top tertile OR 0.33, 95% CI [0.13–0.63], *p* for trend = 0.006, overall *p* = 0.021; per point increase, adjusted OR 0.29, 95% CI [0.13–0.63], *p* = 0.002, Table 3), but not in women. Neither serum levels of total cholesterol (Table S1) nor LDL (Table S2) was associated with sICH risk both in men and in women.

## Sensitivity analysis
Excluding those who received statin treatment before index ischemic stroke event or those who underwent endovascular thrombectomy yielded consistent findings to primary

**Table 1  Baseline characteristics of women compared with men.**

| | Total (n = 957) | Women (n = 329) | Men (n = 628) | P value |
|---|---|---|---|---|
| Age, y (median, IQR) | 68 (59–75) | 70 (60–77) | 66 (58–74) | <0.001 |
| Smoker, n (%) | 298 (31.1%) | 4 (1.2%) | 294 (46.8%) | <0.001 |
| Drinker, n (%) | 116 (12.1%) | 6 (1.8%) | 110 (17.5%) | <0.001 |
| Hypertension, n (%) | 622 (65.0%) | 214 (65.0%) | 408 (65.0%) | 0.981 |
| Diabetes, n (%) | 203 (21.2%) | 62 (18.8%) | 141 (22.5%) | 0.195 |
| Previous stroke, n (%) | 134 (14.0%) | 34 (10.3%) | 100 (15.9%) | 0.018 |
| Hyperlipidemia, n (%) | 324 (33.9%) | 106 (32.2%) | 218 (34.7%) | 0.439 |
| Chronic heart failure, n (%) | 141 (14.7%) | 51 (15.5%) | 90 (14.3%) | 0.628 |
| Ischemic heart disease, n (%) | 111 (11.6%) | 37 (11.2%) | 74 (11.8%) | 0.799 |
| Atrial fibrillation, n (%) | 281 (29.4%) | 134 (40.7%) | 147 (23.4%) | <0.001 |
| Antithrombotics use before stroke, n (%) | | | | 0.343 |
| Antiplatelet, n (%) | 111 (11.6%) | 37 (11.2%) | 74 (11.8%) | |
| Anticoagulation, n (%) | 25 (2.6%) | 12 (3.6%) | 13 (2.1%) | |
| Statin use before stroke, n (%) | 68 (7.9%) | 24 (8.1%) | 44 (7.8) | 0.863 |
| NIHSS (median, IQR) | 6 (3-12) | 8 (4-14) | 6 (3-11) | 0.002 |
| Systolic BP (mmHg) | 149 (133–163) | 151 (132–167) | 148(133–161) | 0.110 |
| TOAST subtype, n (%) | | | | <0.001 |
| Small vessel occlusion | 102 (10.7%) | 32 (9.7%) | 70 (11.2%) | |
| Cardioembolic | 245 (25.6%) | 105 (31.9%) | 140 (22.4%) | |
| Large artery atherosclerosis | 361 (37.7%) | 88 (26.7%) | 273 (43.5%) | |
| Other or underdetermined | 249 (26.0%) | 104 (31.6%) | 143 (22.8%) | |
| Platelet count ($10^9$ /L) | 206 (174–250) | 216 (181–256) | 203 (169–243) | 0.003 |
| Blood glucose, (mmol/L) | 7.00 (5.98–8.65) | 6.84 (6.08–8.34) | 7.12 (5.93–8.90) | 0.444 |
| Total cholesterol, (mmol/L) | 4.56 ± 1.17 | 4.72 ± 1.11 | 4.48 ± 1.20 | 0.002 |
| LDL (mmol/L) | 3.05 ± 1.09 | 3.19 ± 1.12 | 2.98 ± 1.13 | 0.005 |
| Triglyceride (mmol/L) | 1.26 (0.89–1.84) | 1.20 (0.87–1.75) | 1.29 (0.91–1.88) | 0.065 |
| Endovascular thrombectomy | 124(13.0%) | 47(14.3%) | 77(12.3%) | 0.376 |

**Notes.**

Abbreviations: LDL, low-density lipoprotein; NIHSS, National Institutes of Health Stroke Scale; BP, blood pressure.

analyses regarding the association between triglyceride levels and the risk of sICH (Tables 4 and 5). Moderation analysis showed that sex significantly moderated the relationship between serum levels of triglyceride and the risk of sICH ($p < 0.001$).

# DISCUSSION

The current study showed that in men, with increasing serum levels of triglyceride there was a trend towards a lower risk of sICH. However, neither serum levels of total cholesterol nor LDL was associated with sICH in men. In women, none of the lipid profiles was related to sICH after intravenous thrombolysis for acute ischemic stroke.

Several previous studies explored the relationship between baseline lipid profile and intracerebral bleeding risk in AIS population, yielding conflicting findings (*Uyttenboogaart*

**Table 2  Baseline characteristics among patients with and without sICH in men and women.**

| | Men (n = 628) | | | Women (n = 329) | | |
|---|---|---|---|---|---|---|
| | non-sICH (n = 592) | sICH (n = 36) | p value | non-sICH (n = 309) | sICH (n = 20) | p value |
| Age, y (median, IQR) | 66 (58–74) | 71 (67–79) | 0.001 | 70 (60–77) | 75 (68–82) | 0.015 |
| Smoker, n (%) | 282 (47.6%) | 12 (33.3%) | 0.095 | 4 (1.3%) | 0 | >0.999 |
| Drinker, n (%) | 106 (17.9%) | 4 (11.1%) | 0.415 | 6 (1.9%) | 0 | >0.999 |
| Hypertension, n (%) | 382 (64.5%) | 26 (72.2%) | 0.347 | 197 (63.8%) | 17 (85.0%) | 0.091 |
| Diabetes, n (%) | 130 (22.0%) | 11 (30.6%) | 0.230 | 54 (17.5%) | 8 (40.0%) | 0.013 |
| Previous stroke, n (%) | 95 (16.0%) | 5 (13.9%) | 0.731 | 30 (9.7%) | 4 (20.0%) | 0.277 |
| Hyperlipidemia, n (%) | 214 (36.1%) | 4 (11.1%) | 0.004 | 102 (33.0%) | 4 (20.0%) | 0.337 |
| Chronic heart failure, n (%) | 85 (14.4%) | 5 (13.9%) | 0.938 | 46 (14.9%) | 5 (25.0%) | 0.226 |
| Ischemic heart disease, n (%) | 71 (12.0%) | 3 (8.3%) | 0.690 | 32 (10.4%) | 5 (25.0%) | 0.100 |
| Atrial fibrillation, n (%) | 131 (22.1%) | 16 (44.4%) | 0.002 | 118 (38.2%) | 16 (80.0%) | 0.001 |
| Antithrombotics use before stroke, n (%) | 84 (14.2%) | 3 (8.3%) | 0.351 | 45 (14.5%) | 4 (20.0%) | 0.419 |
| Antiplatelet, n (%) | 71 (12.0%) | 3 (8.3%) | | 35 (11.3%) | 2 (10.0%) | |
| Anticoagulation, n (%) | 13 (2.2%) | 0 | | 10 (3.2%) | 2 (10.0%) | |
| Statin use before stroke, n (%) | 41 (7.8%) | 3 (8.6%) | >0.999 | 24 (8.7%) | 0 | 0.387 |
| NIHSS score | 6 (3–11) | 13 (9–17) | <0.001 | 7 (3–13) | 13 (8–18) | 0.001 |
| Stroke subtype, n (%) | | | 0.005 | | | <0.001 |
| Small vessel occlusion | 69 (11.7%) | 1 (2.8%) | | 31 (10.0%) | 1 (5.0%) | |
| Cardioembolic | 126 (21.4%) | 14 (38.9%) | | 89 (28.8%) | 16 (80.0%) | |
| Large atherosclerosis | 254 (43.1%) | 19 (52.8%) | | 86 (27.8%) | 2 (10.0%) | |
| Other or underdetermined | 143 (22.8%) | 2 (5.6%) | | 103 (33.3%) | 1 (5.0%) | |
| Systolic BP (mmHg) | 148 (133–161) | 142 (131–160) | 0.563 | 150 (132–167) | 152 (133–182) | 0.727 |
| Platelet count ($10^9$ /L) | 203 (172–246) | 179 (141–216) | 0.005 | 216 (181–256) | 215 (158–264) | 0.748 |
| Blood glucose (mmol/L) | 7.03 (5.90–8.76) | 8.85 (6.99–11.83) | 0.001 | 6.77 (6.03–8.14) | 8.58 (7.01–11.85) | 0.004 |
| Total cholesterol (mmol/L) | 4.50 ± 1.20 | 4.14 ± 1.03 | 0.081 | 4.73 ± 1.09 | 4.63 ± 1.41 | 0.715 |
| LDL (mmol/L) | 2.98 ± 1.18 | 2.65 ± 0.89 | 0.101 | 3.15 ± 1.06 | 3.34 ± 1.79 | 0.478 |
| Triglyceride (mmol/L) | 1.31 (0.94–1.90) | 0.88 (0.65–1.14) | <0.001 | 1.21 (0.86–1.72) | 1.16 (0.96–2.13) | 0.495 |

**Notes.**

Abbreviations: LDL, low-density lipoprotein; NIHSS, National Institutes of Health Stroke Scale; BP, blood pressure.

et al., 2008). A prospective hospital-based study of 252 AIS patients showed that high admission triglyceride levels were significantly associated with a higher risk of sICH after intravenous r-tPA, while total cholesterol, LDL, and pre-stroke statin use had no influence on the occurrence of sICH (*Uyttenboogaart et al., 2008*). A secondary analysis of the Get With The Guidelines-Stroke (GWTG-Stroke) cohort of thrombolyzed AIS patients showed that serum triglyceride level was modestly associated with sICH. However, neither total cholesterol nor LDL was associated with sICH (*Messé et al., 2013*). A retrospective observational study of 1,066 AIS patients did not detect the associations of sICH with a

**Table 3  Association of lipid profile and the risk of sICH.**

| | Unadjusted OR 95% CI | p-value | Age-NIHSS-adjusted OR 95% CI | p-value | Multivariable OR 95% CI | p-value |
|---|---|---|---|---|---|---|
| **Men** (n = 628) | | | | | | |
| Triglyceride | | 0.008 | | 0.028 | | 0.021 |
| Lowest tertile | Ref | | Ref | | Ref | |
| Medium tertile | 0.39(0.17–0.88) | | 0.41 (0.18–0.95) | | 0.39(0.17–0.91) | |
| Top tertile | 0.29 (0.12–0.71) | | 0.35 (0.14–0.88) | | 0.33(0.13–0.84) | |
| Continuous (per mmol/l increase) | 0.26 (0.12–0.56) | 0.001 | 0.29 (0.13–0.64) | 0.002 | 0.29(0.13–0.63) | 0.002 |
| **Women** (n = 329) | | | | | | |
| Triglyceride | | 0.646 | | 0.605 | | 0.440 |
| Lowest tertile | Ref | | Ref | | Ref | |
| Medium tertile | 0.91 (0.28–2.96) | | 1.08 (0.32–3.64) | | 1.87 (0.23–3.28) | |
| Top tertile | 1.49 (0.52–4.25) | | 1.68 (0.57–4.96) | | 1.88 (0.57–6.23) | |
| Continuous (per mmol/l increase) | 1.18 (0.80–1.74) | 0.417 | 1.19 (0.80–1.76) | 0.386 | 1.19 (0.77–1.84) | 0.435 |

Notes.

In men, adjusted for age, smoker, atrial fibrillation, NIHSS, stroke type, platelet count, and blood glucose ($P < 0.1$ in the univariable analysis). In women, adjusted for age, hypertension, diabetes, atrial fibrillation, NIHSS, stroke type, and blood glucose ($P < 0.1$ in the univariable analysis).

Abbreviations: CI, confidence interval; OR, odds ratio; LDL, low-density lipoprotein; NIHSS, National Institutes of Health Stroke Scale.

**Table 4  Sensitivity analysis limited to those who did not undergo endovascular thrombectomy.**

| | Unadjusted OR 95% CI | p-value | Age-and NIHSS-adjusted OR 95% CI | p-value | Multivariable OR 95% CI | p-value |
|---|---|---|---|---|---|---|
| **Men** (n = 551) | | | | | | |
| Triglyceride | | 0.005 | | 0.022 | | 0.005 |
| Lowest tertile | Ref | | Ref | | Ref | |
| Medium tertile | 0.26 (0.08–0.81) | | 0.30 (0.09–0.94) | | 0.23 (0.07–0.77) | |
| Top tertile | 0.17 (0.05–0.62) | | 0.23 (0.07–0.84) | | 0.16 (0.04–0.59) | |
| Continuous (per mmol/L increase) | 0.16 (0.05–0.51) | 0.002 | 0.18 (0.06–0.58) | 0.004 | 0.17(0.06–0.54) | 0.003 |
| **Women** (n = 282) | | | | | | |
| Triglyceride | | 0.402 | | 0.395 | | 0.175 |
| Lowest tertile | Ref | | Ref | | Ref | |
| Medium tertile | 1.67 (0.36–7.65) | | 1.80 (0.38–8.46) | | 1.46 (0.27–8.07) | |
| Top tertile | 2.63 (0.64–10.82) | | 2.72 (0.64–11.52) | | 4.39 (0.84–22.99) | |
| Continuous (per mmol/L increase) | 1.17 (0.73–1.85) | 0.518 | 1.19 (0.74–1.91) | 0.473 | 1.41 (0.77–2.60) | 0.269 |

Notes.

In men, adjusted for age, smoker, atrial fibrillation, NIHSS, stroke type, platelet count, and blood glucose ($P < 0.1$ in the univariable analysis. In women, adjusted for age, hypertension, diabetes, atrial fibrillation, NIHSS, stroke type, and blood glucose ($P < 0.1$ in the univariable analysis).

Abbreviations: CI, confidence interval; OR, odds ratio; LDL, low-density lipoprotein; NIHSS, National Institutes of Health Stroke Scale.

higher level of LDL (OR 0.96, 95% CI [0.36–2.60], $p = 0.942$), triglyceride (OR 1.74, 95% CI [0.84–3.56], $p = 0.132$), and a lower level of HDL (OR 1.78, 95% CI [0.68–4.65], $p = 0.279$) (*Rocco et al., 2012*). Data from the Stroke Acute Management with Urgent Risk-factor Assessment and Improvement (SAMURAI) rt-PA Registry showed that of 489 enrolled AIS patients (171 women, 70.8 ± 11.6 years old), there were no significant associations between ICH and any lipid levels (*Makihara et al., 2012*). In a hospital-based cohort of consecutive ischemic stroke patients, total cholesterol (OR 0.98, 95% CI [0.98–0.99];

**Table 5 Sensitivity analysis limited to those who did not underwent pre-stroke statin treatment.**

| | Unadjusted OR 95% CI | p-value | Age-and NIHSS-adjusted OR 95% CI | p-value | Multivariable OR 95% CI | p-value |
|---|---|---|---|---|---|---|
| **Men (n = 520)** | | | | | | |
| Triglyceride | | 0.028 | | 0.088 | | 0.171 |
| Lowest tertile | Ref | | Ref | | Ref | |
| Medium tertile | 0.47 (0.20–1.08) | | 0.50 (0.21–1.21) | | 0.54 (0.22–1.33) | |
| Top tertile | 0.30 (0.12–0.79) | | 0.36 (0.13–0.99) | | 0.42 (0.15–1.16) | |
| Continuous (per mmol/l increase) | 0.25 (0.11–0.58) | 0.001 | 0.28 (0.12–0.67) | 0.004 | 0.27 (0.12–0.75) | 0.003 |
| **Women (n = 271)** | | | | | | |
| Triglyceride | | 0.616 | | 0.598 | | 0.260 |
| Lowest tertile | Ref | | Ref | | Ref | |
| Medium tertile | 0.93 (0.28–3.04) | | 1.15 (0.33–3.94) | | 0.68 (0.17–2.76) | |
| Top tertile | 1.54 (0.54–4.45) | | 1.73 (0.58–5.21) | | 2.11 (0.58–7.59) | |
| Continuous (per mmol/l increase) | 1.17 (0.80–1.72) | 0.420 | 1.16 (0.79–1.72) | 0.443 | 1.15 (0.74–1.80) | 0.530 |

**Notes.**

In men, adjusted for age, smoker, atrial fibrillation, NIHSS, stroke type, platelet count, and blood glucose ($P < 0.1$ in the univariable analysis. In women, adjusted for age, hypertension, diabetes, atrial fibrillation, NIHSS, stroke type, and blood glucose ($P < 0.1$ in the univariable analysis).

Abbreviations: CI, confidence interval; OR, odds ratio; LDL, low-density lipoprotein; NIHSS, National Institutes of Health Stroke Scale.

$p = 0.006$) and LDL (OR 95% CI 0.98, [0.97–0.99]; $p = 0.023$) were significantly associated with poor prognosis at 90 days in men, but not in women (*Cuadrado-Godia et al., 2009*). The heterogeneity in the above-mentioned studies may be attributed to a heterogeneous population with different treatment modalities. However, whether the relationship of sICH with lipid profiles may vary by sex has been scarcely reported in the literature. The present study, to the best of our knowledge, is the first to show an association between increasing serum levels of triglyceride and a lower risk of sICH only in men, but not in women, suggesting the potential relationship of sICH with lipid profiles may vary by sex. Even if not separated by gender a meta-analysis showed that the benefits of lipid lowering therapy in prevention of ischemic stroke greatly exceed the risk of ICH. Concern about sICH should not discourage the use of lipid lowering therapy for secondary prevention of ischemic stroke (*Judge et al., 2019*).

Several possible explanations exist for the relationship between increasing triglyceride level and decreased risk of sICH. First, since triglyceride-enriched lipoproteins are accompanied by elevations in factor VII clotting activity and plasminogen activator inhibitor (PAI-1), a higher level of triglyceride may be associated with an imbalance of coagulation over fibrinolysis (*Rosenson & Lowe, 1998*). This hypothesis is supported by a case-control study showing that PAI-1 antigen levels fell by 40% in individuals with a triglyceride-lowering of 20% or more (*Mussoni et al., 1992*). Second, hypotriglyceridemic may be related to the change in viscosity of blood and plasma, which is associated with pathological conditions including hypotension, hypertension, and hemorrhage (*Martini et al., 2006*; *Otto, Richter & Schwandt, 2000*). Third, the proportion of cardioembolic stroke in men was higher than in women (105 (31.9%) *vs.* 140 (22.3%)); thus the paradoxical association between hyper triglyceridemic and sICH may be influenced by this specific distribution of stroke etiology in men (*Amarenco & Steg, 2007*). However, adjusting for

initial stroke severity and subtypes did not change our findings. Therefore, a plausible explanation for the protective role of the so-called "harmful lipid" in male AIS patients is still lacking. Future studies with larger sample sizes are essential to address the prognosis according to etiological subtypes. The reason that high serum levels of triglycerides correlate with a lower risk of sICH in men but not in women remains unclear, which may be attributable to sex differences in lipids metabolism (*Yost et al., 1998*). Further research is required to better elucidate this interesting uncertainty.

### Study strengths and limitations

The strengths of this study included data from multicenter stroke registries with a standard lipid program to assess the lipid profile and central adjudication of the outcome. Our study has several limitations. First, this retrospective study analyzed a small number of patients with considerate heterogeneity, thus it is difficult (or maybe too early) to draw a meaningful conclusion here. Second, this study only included thrombolylzed patients using r-tPA. Thus, our findings could not be generalizable to those who did not receive intravenous r-tPA. Third, although we found a similar proportion of pre-stroke statin use in men and women, information of the use of triglyceride-lowering medications, such as fibrates and omega-3 fatty acids, was unavailable. However, to date, there was no evidence showing the effect of triglyceride-lowering medications on the risk of intracerebral bleeding risk (*Aung et al., 2018*; *Das Pradhan et al., 2022*). Fourth, serum cholesterol and triglyceride levels are not reliable when not measured in a fasting state and may decrease in the acute phase of stroke (*Phuah et al., 2016*). In the current study, lipid levels were measured only once in the fasting state in the acute phase of stroke. Therefore, we do not have an accurate explanation for how temporal lipid levels affect the occurrence of sICH following thrombolysis. Additionally, we were unable to calculate the estimated effect size adjusted for some key unmeasured confounders, such as time from onset to IVT (within 0–3 *vs.* 3–4.5 h) and intensive blood pressure control in the first 24 h (<180/105 mmHg). Lastly, all analyses in the present study are secondary and should be considered hypothesis-generating only.

## CONCLUSIONS

The association between serum levels of triglyceride and sICH may vary by sex. In men, increased triglyceride levels decrease the risk of sICH; in women, this association was lost. Further studies on the biological mechanisms for sex differences in stroke risk associated with triglyceride are needed.

### List of abbreviations

| | |
|---|---|
| **AIS** | acute ischemic stroke |
| **CI** | confidence interval |
| **ECASS** | European Cooperative Acute Stroke Study |
| **IQR** | interquartile range |
| **LDL** | low-density lipoprotein |
| **NINDS** | National Institute of Neurological Disorders and Stroke |

| **NIHSS** | National Institutes of Health Stroke Scale |
| **OR** | odds ratio |
| **r-tPA** | recombinant tissue plasminogen activator |
| **SBP** | systolic blood pressure |
| **TOAST** | The Trial of Org 10172 in Acute Stroke Treatment |

### Funding

The research was sponsored by the Fujian Provincial Key Clinical Specialty of Neurology (No. 0219002). The funders had no role in study design, data collection and analysis, decision to publish, or preparation of the manuscript.

### Grant Disclosures

The following grant information was disclosed by the authors:
The Fujian Provincial Key Clinical Specialty of Neurology: No. 0219002.

### Competing Interests

The authors declare there are no competing interests.

### Author Contributions

- Qilin Yuan conceived and designed the experiments, performed the experiments, analyzed the data, prepared figures and/or tables, authored or reviewed drafts of the article, and approved the final draft.
- Ying Han conceived and designed the experiments, performed the experiments, prepared figures and/or tables, authored or reviewed drafts of the article, and approved the final draft.
- Shuangfang Fang conceived and designed the experiments, performed the experiments, prepared figures and/or tables, authored or reviewed drafts of the article, and approved the final draft.
- Hanhan Lei conceived and designed the experiments, analyzed the data, authored or reviewed drafts of the article, and approved the final draft.
- Huapin Huang conceived and designed the experiments, authored or reviewed drafts of the article, and approved the final draft.
- Huiying Lin conceived and designed the experiments, authored or reviewed drafts of the article, and approved the final draft.
- Xiaomin Wu conceived and designed the experiments, authored or reviewed drafts of the article, and approved the final draft.
- Ronghua Chen conceived and designed the experiments, authored or reviewed drafts of the article, and approved the final draft.
- Zhiting Chen conceived and designed the experiments, authored or reviewed drafts of the article, and approved the final draft.
- Jie Chen conceived and designed the experiments, authored or reviewed drafts of the article, and approved the final draft.

- Hangfeng Li conceived and designed the experiments, authored or reviewed drafts of the article, and approved the final draft.
- Nan Liu conceived and designed the experiments, analyzed the data, authored or reviewed drafts of the article, and approved the final draft.
- Houwei Du conceived and designed the experiments, performed the experiments, analyzed the data, prepared figures and/or tables, authored or reviewed drafts of the article, and approved the final draft.

## Human Ethics

The following information was supplied relating to ethical approvals (i.e., approving body and any reference numbers):

The studies involving human participants were reviewed and approved by the Fujian Medical University Union Hospital Ethics Committees (NO. 2019KY076).

## Data Availability

The raw measurements are available in the Supplemental Files.

## Supplemental Information

Supplemental information for this article can be found online at http://dx.doi.org/10.7717/peerj.17558#supplemental-information.

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
