# Peer review of "Sex difference in the association between triglyceride and intracerebral bleeding risk after intravenous thrombolysis for acute ischemic stroke, a multi-center retrospective study"

_PeerJ, doi:10.7717/peerj.17558_

## Round 0.1 · original submission · Major Revisions

The authors are invited to address the concerns raised by the reviewers.

**Language Note:** The review process has identified that the English language must be improved. PeerJ can provide language editing services - please contact us at [email protected] for pricing (be sure to provide your manuscript number and title). Alternatively, you should make your own arrangements to improve the language quality and provide details in your response letter. – PeerJ Staff

·

Basic reporting

1. The reason for performing the study is unclear.
2. The study’s aim is different between the abstract and the introduction. What is the study's aim?

Experimental design

1. Line 142 to 143. This is a retrospective study. sICH were evaluated by CT or MRI and NIHSS data, how to performed under the condition that “SICH were blindly and centrally adjudicated by an experienced neurologist.”

Validity of the findings

.1. Table 3 shows that increased triglyceride levels are associated with decreased risk of sICH. (The results and conclusion in the abstract are different from the results in the text and table.).
2. Line 180 to 181. The proportion of atrial fibrillation is higher in men than in women. It is wrong. 23.4% in men and 40.7% in women.
3. Line 196. Hypertension did not reach statistical significance.
4. The data description needs to be clearer. Variables associated with sICH older age or younger age, higher blood sugar or lower blood sugar, which type of stroke?

Additional comments

1. In the abstract. The conclusion should be clearer. It should state that increased triglyceride levels increase or decrease the risk of sICH in men.
2. Line 304 to 305. Whether it means that a higher serum triglyceride level is associated with a higher risk of sICH?

Reviewer 2 ·

Basic reporting

The article meet the journal's standard.

Experimental design

This Original primary research was within Aims and Scope of the journal. Research question well defined. Rigorous investigation performed to ethical standard. Methods described with sufficient detail.

Validity of the findings

All underlying data have been provided; they are robust, statistically sound, & controlled. Conclusions are well stated, linked to original research question &

Additional comments

The authors investigated potential sex differences in the association between lipid profile and the risk of sICH in patients with AIS who received intravenous thrombolysis using rt-PA. This work included data from multicenter stroke registries with a standard lipid program to assess the lipid profile and central adjudication of the outcome. The study's meticulous design, involving a sizable and diverse cohort of 957 participants, underscored its robustness. The findings, particularly the observed link between higher serum levels of triglycerides and increased sICH risk in men, offer novel insights into sex-specific factors influencing stroke outcomes. However, I have some comments.

1. This multi-center retrospective study consecutively included AIS patients receiving rt-PA from three teaching hospitals between January 2013 and December 2022. Overall, 957 participants were included. Please provide a flow chart for this study and explain how to handle missing data.

2. Please provide P for trend in the regression analysis showed that serum level of triglyceride was inversely associated with the risk of sICH.

3. A mediation analysis would be better performed to illustrate the mediating role of gender in the impact of blood lipid levels on sICH.

4. Some grammar and writing deficiencies. “Table 3 and Supplemental Tabel 1, 2 shows the associations of lipid profile with the risk of sICH”. “The present study, to the best of our knowledge, is the first to show an association between decreasing serum levels of triglyceride and a higher risk of sICH only in men, but not in women, suggesting the potential relationship of sICH with lipid profiles may vary by sex.” The above statements are slightly incorrect.

5. In the discussion section, it would be beneficial to elaborate on why serum levels of triglycerides added to the risk of sICH in men, whereas in women, this association was not observed.

Reviewer 3 ·

Basic reporting

no comments

Experimental design

no comments

Validity of the findings

no comments

Additional comments

Interesting retrospective review by the authors on sex difference in association with triglycerides and ICH bleeding risk after IV thrombolysis therapy in AIS patients. Well written by the authors and I enjoyed reading.

There are 2 independent and confounding risk factors for hemorrhagic transformation of an AIS following IV thrombolysis that the authors needs to address:
1. Time line: last seen normal to receiving IVT. How many patients received IVT in the 0-3 hours vs 3-4.5 hours in the 2 (male/female) groups. This is important because the longer the time (extended window) there is increase in risk of hemorrhage.
2. Tight blood pressure control in the first 24 hours (<180/105).
Please consider adding these variables, if unable to do so discuss in your limitations.

---

## Round 0.2 · Minor Revisions

The authors are encouraged to address the final remaining issues indicated by the reviewer.

·

Basic reporting

Most of my comments have been revised. However, there are still some issues to clarify.

Experimental design

The design of the study is acceptable.

Validity of the findings

Results
1. Line 187 to 188. "The proportion of atrial fibrillation was higher in men than in women." It is wrong. 23.4% of men and 40.7% of women. Atrial fibrillation in women is higher than in men.

2. Table 4 and Table 5 show the data excluding patients who underwent endovascular thrombectomy and those who used statin before stroke. The patient number is the same as in Table 3. Table 3 includes data on patients who used statins before stroke and who received an endovascular thrombectomy.


Discussion
3. There are some mistakes between lines 286 and 289, please correct them.

Additional comments

Accept after revision.

---

## Round 0.3 · accepted · Accept

The authors have satisfactorily addressed all the comments raised by the reviewers.

·

Basic reporting

No comment.

Experimental design

No comment.

Validity of the findings

No comment.

Additional comments

Accept.

Reviewer 2 ·

Basic reporting

Clear, sufficient field background,Professional article structure, figures, tables

Experimental design

Original primary research within Aims and Scope of the journal

Validity of the findings

All underlying data have been provided; they are robust, statistically sound, & controlled.

Additional comments

Thank you for inviting me to review the article again. I have no further suggestions for the revised manuscript, except for one: please add the corresponding reference to the statement in the discussion section on page 13, "The Guidelines-Stroke (GWTG-Stroke) cohort of thrombolyzed AIS patients showed that serum triglyceride level was modestly associated with sICH."